# Total *Sn*-2 Palmitic Triacylglycerols and the Ratio of OPL to OPO in Human Milk Fat Substitute Modulated Bile Acid Metabolism and Intestinal Microbiota Composition in Rats

**DOI:** 10.3390/nu15234929

**Published:** 2023-11-26

**Authors:** Lin Zhu, Shuaizhen Fang, Hong Zhang, Xiangjun Sun, Puyu Yang, Jianchun Wan, Yaqiong Zhang, Weiying Lu, Liangli Yu

**Affiliations:** 1Institute of Food and Nutraceutical Science, School of Agriculture and Biology, Shanghai Jiao Tong University, Shanghai 200240, China; julia_1113@sjtu.edu.cn (L.Z.); fangsz9918@sjtu.edu.cn (S.F.); xjsun@sjtu.edu.cn (X.S.); yangpuyuallen@sjtu.edu.cn (P.Y.); weiying.lu@sjtu.edu.cn (W.L.); 2Wilmar (Shanghai) Biotechnology Research & Development Center Co., Ltd., Shanghai 200137, China; zhanghongsh@cn.wilmar-intl.com (H.Z.); wanjianchun@cn.wilmar-intl.com (J.W.); 3Department of Nutrition and Food Science, University of Maryland, College Park, MD 20742, USA; lyu5@umd.edu

**Keywords:** *sn*-2 palmitic triacylglycerols, human milk fat substitute, BA metabolism, FXR, TGR5, gut microbiota

## Abstract

In this study, the impact of *sn*-2 palmitic triacyclglycerols (TAGs) in combination with their ratio of two major TAGs (1-oleoyl-2-palmitoyl-3-linoleoylglycerol (OPL) to 1,3-dioleoyl-2-palmitoylglycerol (OPO)) in human milk fat substitute (HMFS) on bile acid (BA) metabolism and intestinal microbiota composition was investigated in newly-weaned Sprague–Dawley rats after four weeks of high-fat feeding. Compared to those of control group rats, HMFS-fed rats had significantly increased contents of six hepatic primary BAs (CDCA, αMCA, βMCA, TCDCA, TαMCA and TβMCA), four ileal primary BAs (UDCA, TCA, TCDCA and TUDCA) and three secondary BAs (DCA, LCA and ωMCA), especially for the HMFS with the highest *sn*-2 palmitic acid TAGs of 57.9% and OPL to OPO ratio of 1.4. Meanwhile, the inhibition of ileal FXR-FGF15 and activation of TGR5-GLP-1 signaling pathways in HMFS-fed rats were accompanied by the increased levels of enzymes involved in BA synthesis (CYP7A1, CYP27A1 and CYP7B1) in the liver and two key thermogenic proteins (PGC1α and UCP1) in perirenal adipose tissue, respectively. Moreover, increasing *sn*-2 palmitic TAGs and OPL to OPO ratio in HMFS also altered the microbiota composition both on the phylum and genus level in rats, predominantly microbes associated with bile-salt hydrolase activity, short-chain fatty acid production and reduced obesity risk, which suggested a beneficial effect on host microbial ecosystem. These observations provided important nutritional evidence for developing new HMFS products for infants.

## 1. Introduction

Fatty acids are the main components of triacylglycerols (TAGs), which are the major components of dietary lipids. Many previous studies have shown that varying the saturation degree or acyl chain length of fatty acids in the dietary lipids may affect the host physiology states [1,2]. Recently, it was found that the fatty acid stereo-distribution in TAGs could affect the nutritional function of dietary lipids. *Sn*-2 palmitic TAGs are a group of structured TAGs with a higher proportion of palmitic acid (PA) esterified at the *sn*-2 position, such as 1,3-dioleoyl-2-palmitoylglycerol (OPO) and 1-oleoyl-2-palmitoyl-3-linoleoylglycerol (OPL), which is similar to the structure of TAG present in human breast fat [3]. Several clinical trials and animal studies suggested that *sn*-2 palmitic TAGs might have better digestion and absorption properties, along with some other beneficial physiological functions, such as improved calcium absorption, bone matrix quality, consistency of stools and sleeping time [4]. Thus, the results of these studies greatly support the potential application of *sn*-2 palmitic TAGs in human milk fat substitute (HMFS) design and infant formula formulations [5]. Moreover, our recent studies found that both *sn*-2 palmitic TAGs and their TAG composition could affect rats’ growth and lipid metabolism [6,7]. Increasing the total *sn*-2 palmitic TAGs and the ratio of OPL to OPO in HMFS could reduce body weight gain and excessive accumulation of lipid droplets in the liver and perirenal adipose tissue, and improve serum lipid indicators in rats after 4 weeks of high-fat feeding. However, the underlying reason has yet to be elucidated.

In the past decades, bile acid (BA) has been recognized as important signaling molecules that influence the lipid metabolism of the host. It is known that primary BAs are synthesized in hepatocytes from cholesterol and released into the intestine to facilitate the absorption of dietary lipids [8]. In addition, primary BAs could be further metabolized into secondary BAs by gut microbiota enzymes in the intestine [9]. It is clear that gut microbiota can regulate BA homeostasis through two main BA signaling receptors, farnesoid X receptor (FXR) and G protein-coupled receptor 5 (TGR5) [10]. On the other hand, BAs can reshape gut microbiota composition through the direct modulation of bile-sensitive and bile-metabolizing bacteria [11]. Therefore, there is a dynamic interplay between BAs and gut microbiota, which further regulates BA homeostasis, lipid metabolism, and energy harvest [12]. To the best of our knowledge, no relevant in vivo studies have been reported on whether and how *sn*-2 palmitic TAGs and the ratio of OPL to OPO may alter the BA metabolism and gut microbiota composition.

In the present study, it was hypothesized that *sn*-2 palmitic TAGs and the OPL to OPO ratio in the HMFS could affect BA metabolism and the gut microbiota response. To test the hypothesis, newly weaned Sprague–Dawley (SD) rats were fed with the HMFS with different *sn*-2 palmitic TAGs and OPL to OPO ratios. The hepatic and ileal BA composition in HMFS-fed rats was first measured, in comparison with those of the control group rats. The influence of key proteins involved in BA synthesis in the liver, hepatic FXR-small heterodimer partner (SHP), intestinal FXR-fibroblast growth factor 15 (FGF15) and TGR5-glucagon-like peptide-1 (GLP-1) signaling pathways were also investigated. In addition, the gut microbiota composition may further help us understand the influence of HMFS on metabolic alternations in rats.

## 2. Materials and Methods

### 2.1. Materials

BA standards (cholic acid (CA), chendoxycholic acid (CDCA), ursodeoxycholic acid (UDCA), α muricholic acid (αMCA), β muricholic acid (βMCA), tauro cholic acid (TCA), tauro chendoxycholic acid (TCDCA), tauro ursodeoxycholic acid (TUDCA), tauro α muricholic acid (TαMCA), tauro β muricholic acid (TβMCA), deoxycholic acid (DCA), lithocholic acid (LCA), ω muricholic acid (ωMCA), tauro deoxycholic acid (TDCA) and tauro ω muricholic acid (TωMCA)) were obtained from Sigma-Aldrich (St. Louis, MO, USA). LC-grade acetonitrile, methanol and formic acid were supplied by Merck (Darmstadt, Germany). All the other reagents were bought from Aladdin Co., Ltd. (Shanghai, China).

### 2.2. Animals and Diets

All animal procedures were performed according to guidelines for the use of laboratory animals, which were approved by the Ethics Committee (EC) of Shanghai Jiao Tong University. Since this work is a continuation of our previous work, the detailed animal feeding and grouping were the same as those previously reported [6]. Briefly, forty male 3-week-old SD rats were acclimatized to the environmentally standard condition for 1 week. The rats were then randomly divided into four groups, which were fed with four different dietary lipids for 4 weeks, respectively. Rats fed with the control fat (*sn*-2 palmitic acid level of 15.5%, OPL to OPO ratio of 0.4) were named as the control fat (CF) group. Meanwhile, the other three groups of rats were fed with the HMFS1 (*sn*-2 palmitic acid level of 54.4%, OPL to OPO ratio of 0.3), HMFS2 (*sn*-2 palmitic acid level of 60.0%, OPL to OPO ratio of 0.9) and HMFS3 (*sn*-2 palmitic acid level of 57.9%, OPL to OPO ratio of 1.4), respectively. The detailed chemical compositions of four dietary lipids is listed in Appendix A, and the detailed diet composition is listed in Appendix A.

### 2.3. Collection of Animal Samples

Rats were anesthetized with carbon dioxide prior to execution. Liver, perirenal fat, ileum, as well as intestinal contents of rats were harvested and stored at −80 °C for further analysis.

### 2.4. Hepatic and Ileal BA Contents

The hepatic and ileal BAs were measured on the ultrahigh performance liquid chromatography/triple quadrupole mass spectrometer (UHPLC/TQ-MS), according to our previously published literature [6].

### 2.5. Protein Level Analysis

The liver (110 mg), ileum (80 mg) and adipose tissue (50 mg) were rinsed with pre-cooled PBS (pH = 7.4) to remove the residual blood, respectively. Then, the tissue was homogenized and centrifuged at 5000× *g* for 10 min in pre-cooled PBS, and the supernatant was taken for testing. The levels of cholesterol 7α-hydroxylase (CYP7A1), sterol 12α-hydroxylase (CYP8B1), sterol-27-hydroxylase (CYP27A1), oxysterol 7α-hydroxylase (CYP7B1), FXR and SHP in liver, FXR, FGF15, TGR5 and GLP-1 in ileum, and peroxisome proliferators-activated receptor γ coactivator l-alpha (PGC1α) and uncoupling protein 1 (UCP1) in perirenal adipose tissue of SD rats were measured using enzyme-linked Immunosorbent Assay (ELISA) kits (LifeSpan BioSciences Inc., Seattle, WA, USA).

### 2.6. DNA Extraction from Intestinal Contents

The E.Z.N.A.^®^ soil DNA Kit was used to extract microbial DNA from the intestinal contents of rats [13]. DNA concentration and purity were obtained by determining the absorption at 260 and 280 nm using the NanoDrop spectrophotometer (Thermo Scientific, Wilmington, DE, USA).

### 2.7. PCR Amplification and Sequencing

The PCR methods employed were the same as previously reported [14]. Briefly, the V3-V4 hypervariable regions of the bacteria 16S rRNA gene were amplified by PCR system (GeneAmp 9700, ABI, USA). Quantus Fluorometer E6150 (Promega, Madison, WI, USA) was used to quantify the product. Illumina Miseq platform (Illumina, San Diego, CA, USA) was used to perform high-throughput sequencing.

### 2.8. Bioinformatics Analysis

FLASH software was used to obtain the final effective tags. Uparse software was used to cluster the effective tags, and the sequences were clustered into Operational taxonomic units (OTUs) at 97% similarity level. The Mothur software and bacterial Silva 106 database were used to carry out basic local alignment search tools (BLASTs) of taxonomic classification down to the phylum and genus level, and calculate Shannon, Simpson, ACE and Chao indices.

### 2.9. Statistical Analyses

One-way ANOVA and Tukey’s post hoc multiple comparisons were employed to identify differences in means between groups (*p* < 0.05 or 0.01). SPSS was used to analyze statistics. Spearman’s correlation analysis was used to conduct the correlation analysis. R studio was used to perform the principal component analysis (PCA).

## 3. Results

### 3.1. Hepatic and Ileal BA Contents

The hepatic and ileal BA contents in CF and HMFS-fed rats were first measured in this study. Based on the PCA observations, the BA profiles among the CF and HMFS-fed rat groups were well differentiated, both for the hepatic (Appendix A) and ileal BAs (Appendix A), which indicated that HMFS may have some influence on the BA profiles in rats. Specifically, a total of 15 BAs in the liver was found in rats, including 10 primary BAs and 5 secondary BAs (Table 1). Compared with those of CF-fed rats, HMFS1-fed rats showed significantly higher contents of six primary BAs, including CDCA, αMCA, βMCA, TCDCA, TαMCA and TβMCA contents (*p* < 0.05). Meanwhile, HMFS3-fed rats had the highest contents of these six primary BAs among the three HMFS-fed rat groups (*p* < 0.05). Moreover, there were also significant differences in TCA in the HMFS3-fed rat group compared to those of the other two HMFS-fed rat groups.

As for the ileal BAs, there were also significant changes in the contents of four primary BAs and one secondary BA between the CF-fed and HMFS1-fed rat groups (Table 2). Compared to those of CF-fed rats, HMFS1-fed rats had significantly higher contents of UDCA, LCA, TCA, TCDCA and TUDCA (*p* < 0.05), suggesting that the increase of *sn*-2 palmitic acid content from 15.5% to 54.4% in HMFS could significantly increase the ileal BA contents, especially for the tauro-conjugated BAs (TCA, TCDCA and TUDCA). Increasing the OPL to OPO ratio in HMFS from 0.3 (HMFS1) to 0.9 (HMFS2) could increase the contents of these five ileal BAs in rats compared to those of HMFS1-fed rats. Moreover, DCA and ωMCA are two other secondary BAs, which are synthesized from CA and βMCA via gut microbiota, respectively [15]. As shown in Table 2, the contents of DCA and ωMCA were the highest for HMFS3-fed rats (*p* < 0.05). Meanwhile, no significant differences in DCA and ωMCA contents were found among CF-fed and HMFS1-fed rats.

### 3.2. Protein Levels Related to BA Metabolism

In this study, the levels of four proteins related to BA synthesis were measured (Figure 1A–D). The statistically higher CYP7A1 level was observed for HMFS1-fed rats (16.41 pg/mL) compared to that of CF-fed rats (12.39 pg/mL). Meanwhile, the CYP7A1 level of HMFS2 and HMFS3-fed rats was further significantly increased to 18.79 pg/mL and 22.26 pg/mL, respectively, compared to that of HMFS1-fed rats (*p* < 0.05). There was no significant difference between the CF-fed and HMFS1-fed rats and among three HMFS-fed rats regarding their CYP8B1 level (*p* > 0.05). In addition, compared to that of CF-fed rats, three groups of HMFS-fed rats all exhibited a significant increase in levels of CYP27A1 and CYP7B1 (*p* < 0.05), with the highest levels also shown in HMFS3-fed rats (23.73 pg/mL and 23.65 pg/mL).

Moreover, compared to those of CF-fed rats, three groups of HMFS-fed rats all exhibited a significantly increased level of hepatic FXR and SHP (*p* < 0.05), with the highest protein levels shown in HMFS3-fed rats (455.29 pg/mL and 9.46 ng/mL) (Figure 1E,F). Meanwhile, HMFS1-fed rats had a significantly decreased level of ileal FXR by 18.70%, compared to that of CF-fed rats (*p* < 0.05). When the OPL to OPO ratio in HMFS increased from 0.3 to 0.9 and 1.4, the ileal FXR level in HMFS2-fed and HMFS3-fed rats were further decreased by 12.75% and 16.27% in comparison to that of HMFS1-fed rats (*p* < 0.05) (Figure 1G). The ileal FGF15 level showed a similar tendency to that of the ileal FXR level (Figure 1H).

As shown in Figure 1I,J, HMFS1-fed rats displayed significantly higher ileal TGR5 and GLP-1 levels in comparison with those of CF-fed rats (*p* < 0.05). In addition, HMFS3-fed rats had the highest ileal TGR5 and GLP-1 levels (13.12 nmol/L and 16.41 nmol/L) among the three HMFS-fed rat groups (*p* < 0.05). The levels of two key thermogenic proteins, including PGC1α and UCP1 were also measured. It was observed that three HMFS-fed rat groups significantly upregulated these protein levels compared with those of the CF-fed rat group, no matter for PGC1α or UCP1 (Figure 1K,L). The highest PGC1α and UCP1 levels were also shown in HMFS3-fed rats (375.82 nmol/L and 1757.24 pg/mL).

### 3.3. Gut Microbiota Community

#### 3.3.1. α-Diversity of Gut Microbiota

In this study, we first evaluated the richness and diversity of gut microbiota in CF-fed and HMFS-fed rats by calculating four α-diversity indices, which have an important role in maintaining the dynamic balance of the intestinal micro-ecosystem [16,17]. Compared to that of CF-fed rats, HMFS1-fed rats had significantly higher ACE and Shannon indices (*p* < 0.05). Meanwhile, a further statistically significant increase in Chao, ACE and Shannon indices was shown in HMFS3-fed rats in comparison to those of HMFS1-fed and HMFS2-fed rats (*p* < 0.05) (Figure 2). Chao and ACE are indices used to reflect the abundance of OTUs in gut microbiota, while Shannon is an index for evaluating the diversity of OUTs in gut microbiota [18].

#### 3.3.2. Relative Abundance of Gut Microbiota on Phylum Level

As shown in Figure 3A, the histogram of relative phylum abundance demonstrates that *Firmicutes* and *Bacteroidetes* comprise the main dominant bacteria regardless of CF-fed and HMFS-fed rats. However, the relative abundance of *Firmicutes* in the ileum of HMFS-fed rats was significantly decreased than that of CF-fed rats. Meanwhile, the variation trend in relative abundance of *Bacteroides* among CF-fed and HMFS-fed rat groups was just the opposite. Therefore, the ratio of *Firmicutes* to *Bacteroidetes* (F/B) was calculated to be 12.47 (CF group), 1.57 (HMFS1 group), 1.67 (HMFS2 group) and 1.72 (HMFS3 group), respectively (Appendix A). Furthermore, compared to those of CF-fed rats, HMFS-fed rats also had an increased abundance of *Actinobacteria* and *Verrucomicrobia*. The statistically highest abundance of these two phyla was shown in HMFS3-fed rats (*p* < 0.05). Although the relatively low abundance of *Proteobacteria* was shown for all four experimental groups (0.5–1.5%), a significantly increased abundance was also shown in HMFS-fed rats, compared to that of CF-fed rats.

#### 3.3.3. Relative Abundance of Gut Microbiota on Genus Level

At the genus level, 16 genera with a mean relative abundance > 1% were classified in CF-fed and HMFS-fed rats. *Akkermansia*, *Lachnospiraceae*, *Lactobacillus* and *Clostridium* were the dominant genera with significant differences in relative abundance among four experimental groups (Figure 3B). Compared to that of CF-fed rats (10.76%), the relative abundance of *Akkermansia* increased significantly to 22.42, 25.30 and 35.95% for HMFS1-fed, HMFS2-fed and HMFS3-fed rats, respectively (*p* < 0.05). In addition, the enrichment of *Lachnospiraceae*, *Clostridium*, *Blautia*, *S24-7* and *Oscillospira* were evidently increased and the abundance of *Lactobacillus*, *Dorea* and *Bacteroides* was also significantly decreased in three HMFS-fed groups, compared to those of the CF-fed group (*p* < 0.05). Meanwhile, significant differences among CF-fed and HMFS-fed rats were also shown in *Lactobacillus*, *Clostridium*, and *Bacteroides*, not in *Bifidobacterium* (Figure 3B). Meanwhile, HMFS3-fed rats had the highest relative abundance of *Clostridium* (11.65%) and lowest relative abundance of *Lactobacillus* and *Bacteroides* (10.94% and 0.34%) among the three HMFS-fed rat groups.

### 3.4. Correlation Analysis

The Spearman correlation analysis was first used to analyze the correlations between hepatic BAs and proteins in the liver, which are involved in BA metabolism (Figure 4A). Hepatic CA content was positively correlated with the CYP7A1 and CYP8B1 levels (CA vs. CYP7A1, r = 0.67, *p* < 0.05; CA vs. CYP8B1, r = 0.79, *p* < 0.05), and hepatic CDCA content was positively correlated with the CYP27A1, CYP7B1, hepatic FXR and SHP levels (CDCA vs. CYP27A1, r = 0.73, *p* < 0.05; CDCA vs. CYP7B1, r = 0.79, *p* < 0.05; CDCA vs. FXR, r = 0.70, *p* < 0.05; CDCA vs. SHP, r = 0.68, *p* < 0.05). In addition, significantly positive correlations between the CYP27A1 level and αMCA or βMCA were also observed. The association between ileal BAs and ileal FXR, FGF15, TGR5 and GLP-1 is also shown in Figure 4B. The significantly negative correlations between TCDCA or TUDCA and FXR or FGF15 in the ileum could be found (r = −0.69 for TCDCA vs. FXR, *p* < 0.05; r = −0.70 for TCDCA vs. FGF15, *p* < 0.05; r = −0.44 for TUDCA vs. FXR, *p* < 0.05; r = −0.55 for TUDCA vs. FGF15, *p* < 0.05). UDCA content was observed to be negatively correlated with the ileal FXR level (r = −0.45 for UDCA vs. FXR, *p* < 0.05). In addition, the contents of LCA and DCA were positively correlated with TGR5 and GLP-1 levels (r = 0.54 for LCA vs. TGR5, *p* < 0.05; r = 0.41 for LCA vs. GLP-1, *p* < 0.05; r = 0.64 for DCA vs. TGR5, *p* < 0.05; r = 0.47 for DCA vs. GLP-1, *p* < 0.05).

The correlations among different proteins were then analyzed. Significantly negative correlations between ileal FXR or FGF15 and CYP7A1, CYP27A1 and CYP7B1 were observed (ileal FXR vs. CYP7A1, r = −0.69, *p* < 0.05; ileal FXR vs. CYP27A1, r = −0.82, *p* < 0.01; ileal FXR vs. CYP7B1, r = −0.75, *p* < 0.01; FGF15 vs. CYP7A1, r = −0.65, *p* < 0.05; FGF15 vs. CYP27A1, r = −0.84, *p* < 0.01; FGF15 vs. CYP7B1, r = −0.69, *p* < 0.01) (Figure 4C). The TGR5 and GLP-1 levels were positively correlated with CYP7B1, respectively (TGR5 vs. CYP7B1, r = 0.65, *p* < 0.05; GLP-1 vs. CYP7B1, r = 0.63, *p* < 0.05). In addition, the significantly positive correlations among PGC1α, UCP1, TGR5 and GLP-1 were also observed (PGC1α vs. TGR5, r = 0.70, *p* < 0.01; PGC1α vs. GLP-1, r = 0.74, *p* < 0.01; UCP1 vs. TGR5, r = 0.80, *p* < 0.01; UCP1 vs. GLP-1, r = 0.86, *p* < 0.01).

Last, the relative abundances of gut microbiota were shown to have some significant correlations with the ileal tauro-conjugated and secondary BAs (Figure 4D). For example, *Clostridium* was positively correlated with the levels of three secondary BAs (LCA, DCA and ωMCA) (r = 0.62 for *Clostridium* vs. LCA, *p* < 0.05; r = 0.83 for *Clostridium* vs. DCA, *p* < 0.05; r = 0.70 for *Clostridium* vs. ωMCA, *p* < 0.05), while the contents of tauro-conjugated BAs (TCA, TCDCA and TUDCA) were negatively correlated with the relative abundance of *Lactobacillus* and *Bacteroides* (r = −0.59 for TCA vs. *Lactobacillus*, *p* < 0.05; r = −0.55 for TCDCA vs. *Lactobacillus*, *p* < 0.05; r = −0.44 for TUDCA vs. *Lactobacillus*, *p* < 0.05; r = −0.45 for TCA vs. *Bacteroides*, *p* < 0.05; r = −0.73 for TCDCA vs. *Bacteroides*, *p* < 0.05; r = −0.46 for TUDCA vs. *Bacteroides*, *p* < 0.05). In addition, the ileal TCA, TCDCA and TUDCA contents were positively correlated with the relative abundances of *Akkermansia* and *Oscillospira*, and negatively correlated with the relative abundance of *Dorea*. Significant positive correlations between *S24-7* relative abundance and TCDCA or TUDCA were also observed.

## 4. Discussion

To clarify the influence of *sn*-2 palmitic TAGs and OPL to OPO ratio in HMFS on BA metabolism and gut microbiota more clearly, a schematic diagram outlining the interrelation among the main BAs, key proteins involved in BA metabolism and BA-associated gut microbiota in rats is shown in Figure 5, which presents an original hypothesis based on all the results obtained in this study.

First, it was found that increasing the *sn*-2 palmitic TAGs and OPL to OPO ratio in HMFS could significantly increase the contents of six primary BAs in rat liver (CDCA, αMCA, βMCA, TCDCA, TαMCA and TβMCA) (*p* < 0.05) (Table 1). It is known that CDCA, αMCA and βMCA can be conjugated with taurine to form these three tauro-conjugated BAs (TCDCA, TαMCA and TβMCA) in hepatocytes, respectively. Therefore, an increased trend of these six primary BAs in HMFS-fed rats is reasonable. Meanwhile, the tauro-conjugated BAs (TCA, TCDCA and TUDCA) can be secreted from hepatocytes and metabolized into secondary BAs via gut microbiota [19]. Therefore, HMFS-fed rats also demonstrated a significant increase in ileal BA levels, especially for the tauro-conjugated and secondary BAs, and the increased trend was positively correlated with the *sn*-2 palmitic TAGs and OPL to OPO ratio in HMFS (Table 2).

Meanwhile, it is known that the primary BAs are synthesized in the liver through two different pathways: the classical and alternative pathways. The classical pathway accounts for at least 75% of BA production, initiated by CYP7A1 and subsequently by CYP8B1. CYP7A1 is the rate-limiting enzyme and determines the produced BA contents. The alternative pathway is initiated by CYP27A1 to form 27-hydroxycholesterol, which is further hydroxylated by CYP7B1 [20]. The classical synthesis pathway generates both CDCA and CA, while the alternative pathway predominantly generates CDCA. The proportion between these two primary BAs is determined by CYP8B1, which is required for CA synthesis [21]. Therefore, our observations that increasing the *sn*-2 palmitic TAGs and OPL to OPO ratio in HMFS could increase the levels of CYP7A1, CYP27A1 and CYP7B1 (Figure 1A,C,D) possibly explained the increased contents of six hepatic primary BAs in the tested rats (Table 1). Moreover, compared to that of CF-fed rats, a significantly increased hepatic CDCA content was shown in HMFS-fed rats, but not for the CA content. This might be due to the level of CYP8B1 required for CA synthesis not being significantly influenced by HMFS, no matter for the *sn*-2 palmitic TAGs or OPL to OPO ratio in HMFS (Figure 1B).

Moreover, FXR has been identified as a BA-activated nuclear receptor that regulates the homeostasis of BAs, which is highly expressed both in the liver and ileum of rats. The activation of hepatic and ileal FXR, and their downstream SHP and FGF15 signaling could suppress BAs synthesis in both classical and alternative pathways. Meanwhile, a more potent inhibition effect was observed for ileal FXR, compared to that of hepatic FXR [10,22]. In this study, it was found that increasing the *sn*-2 palmitic TAGs and OPL to OPO ratio in HMFS could activate the hepatic FXR-SHP signaling pathway, while inhibiting the ileal FXR-FGF15 signaling pathway in rats (Figure 1E–H). Significantly negative correlations between ileal FXR and CYP7A1, CYP27A1 and CYP7B1 were observed, while no significant correlations were observed between hepatic FXR and BAs synthesis enzymes in this study (Figure 4C). Moreover, CDCA is known as a potent agonist for hepatic FXR [23], which could explain its positive correlation with hepatic FXR and its downstream SHP (Figure 4A). Huang et al. found that theaflavin could dramatically decrease the ileal FXR and FGF15 levels and increase the TCDCA and TUDCA contents in the ileum of mice, since TCDCA and TUDCA acted in an antagonistic manner on ileal FXR [19]. Meanwhile, UDCA was also considered to be an antagonist of ileal FXR receptor [24]. The correlation results in this study also confirmed these previous observations (Figure 4B).

Meanwhile, the ileal TGR5 is a type of BA responsive cellular surface receptor, and TGR5 signaling in enteroendocrine L-cells could further induce the secretion of GLP-1 [25,26]. In addition, the contents of LCA and DCA were found to be positively correlated with TGR5 and GLP-1 levels, which might be due to the ability of TGR5 to bind secondary BAs, such as LCA and DCA, which are the most potent agonists for TGR5 [27,28]. Meanwhile, it has also been reported that the activation of TGR5 by BAs could trigger an increase in energy expenditure and protect against diet-induced obesity in the host [29]. The significant positive correlations among PGC1α, UCP1, TGR5 and GLP-1 might also indicate that increasing the *sn*-2 palmitic TAGs and OPL to OPO ratio in HMFS could significantly upregulate the energy metabolism of rats by activating TGR5-GLP-1 signaling (Figure 4C). These results may also support our previous observations that HMFS-fed rats had decreased body weight gain, and less fat accumulation in the liver and perirenal adipose tissue, compared to that of the CF-fed rats [6].

Lastly, the *sn*-2 palmitic TAGs and OPL to OPO ratio in HMFS were also found to exhibit some influence on gut microbiota in rats, which could further reveal the possible reasons for the changes in BAs profiles and levels of proteins involved in BA metabolism in rats. It is known that the tauro-conjugated BAs are released into the intestine of rats and then de-conjugated by bile salt hydrolase (BSH)-active bacteria to form unconjugated BAs (e.g., CA, CDCA and βMCA), including *Lactobacillus*, *Clostridium*, *Bifidobacterium* and *Bacteroides*. Subsequently, CA and CDCA can be 7-dehydroxylated into secondary BAs (e.g., DCA, LCA) via the 7a-dehydroxylation of *Clostridium*, and βMCA can be 6β-epimerized into ωMCA via the 6β-epimerization of *Clostridium* [30,31]. Our results showed that increasing *sn*-2 palmitic TAGs and OPL to OPO ratio in HMFS could increase the relative abundance of *Clostridium* and decrease the relative abundance of *Lactobacillus* and *Bacteroides* in rats (Figure 3B). Meanwhile, the relative abundances of *Lactobacillus* and *Bacteroides* were significantly negatively correlated with the ileal tauro-conjugated BAs (TCA, TCDCA and TUDCA), and the relative abundance of *Clostridium* was significantly positively correlated with the secondary BAs (LCA, DCA and ωMCA) (Table 2, Figure 4D). In a previous study, Jia and others also found that tauro-conjugated BAs had a significant negative correlation with *Lactobacillus* and *Bacteroides*, while secondary BAs had a significant positive correlation with *Clostridium* [32].

Meanwhile, the gut microbiota results in this study also provided some important evidence for elaborating the influence of *sn*-2 palmitic TAGs and OPL to OPO ratio in HMFS on the lipid metabolism of rats, which was found in our previously published literature [6]. *Firmicutes* and *Bacteroidetes* have an important responsiveness to dietary fat, and mice fed with high-fat diets have an increased abundance of *Firmicutes* and decreased abundance of *Bacteroidetes*. Therefore, the increased F/B ratio in gut microbiota has been observed to be positively related to the risk of obesity [33,34]. Zou et al. found that mice fed with inulin had reduced obesity, and meanwhile, they had a decreased F/B ratio in gut microbiota [35]. Therefore, the significantly decreased F/B ratio observed in three HMFS-fed rat groups indicated the reduced risk of obesity in rats (Appendix A), although they were also fed with high-fat diets. Meanwhile, *Proteobacteria* and *Actinobacteria* are known to reduce lipid accumulation in hepatocytes, which could improve hepatic steatosis caused by excessive fat deposition [36]. The relative abundances of gut microbiota at the genus level corresponded well with those at the phylum level. Increasing the *sn*-2 palmitic TAGs and OPL to OPO ratio in HMFS significantly increased the relative abundance of *Akkermansia*, especially for the HMFS3-fed rats (Figure 3B). A previous study also showed that polyphenol-rich cranberry extract protected mice from diet-induced obesity, possibly associated with an increased relative abundance of *Akkermansia* within the gut of mice [37]. In addition, Rao et al. showed that the *Akkermansia* treatment efficiently increased lipid oxidation in high-fat and high-cholesterol diet-fed obese mice, and reversed metabolic dysfunction-associated fatty liver disease in rats [38]. Therefore, the increased relative abundance of *Akkermansia* also supported the changes in relative abundances of *Firmicutes* and *Bacteroidetes* (Figure 3A) and upregulated levels of two energy metabolism-related proteins (PGC1α and UCP1) (Figure 1K,L) in rats, possibly indicating the reduced risk of obesity and increased lipid oxidation in HMFS-fed rats. Meanwhile, some clinical studies also suggested that lower abundances of *Blautia* and *Oscillospira* were commonly observed in obese individuals [39,40], which further indicated the reduced risk of obesity in HMFS-fed rats and was also consistent with the observed change in *Akkermansia* relative abundance. The enrichment of *Lachnospiraceae*, *S24-7* and *Verrucomicrobia*, and decreased abundance of *Dorea* could increase the amount of SCFA in the intestine, resulting in the improved intestinal health of rats [41,42].

## 5. Conclusions

In conclusion, this study showed that *sn*-2 palmitic TAGs and the OPL to OPO ratio in human milk fat substitute exerted some influence on BA metabolism through FXR- and TGR5-mediated signaling pathways in newly weaned SD rats. Moreover, it also altered the richness and diversity of gut microbiota and the microbiota composition in rats, mainly related to BA metabolism, SCFA production and reduced obesity risk. These results provide supporting evidence regarding the nutritional regulatory role of *sn*-2 palmitic TAGs and TAG composition in dietary lipids.

## Figures and Tables

**Figure 1 nutrients-15-04929-f001:**
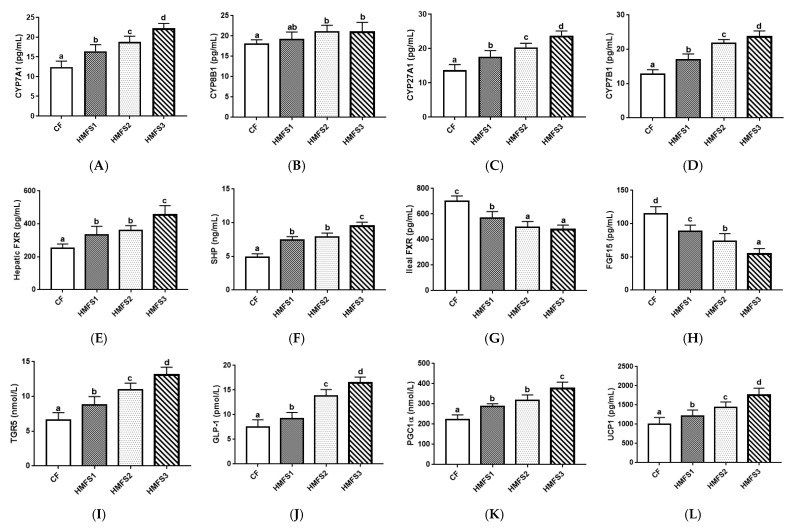
The levels of CYP7A1 (**A**), CYP8B1 (**B**), CYP27A1 (**C**), CYP7B1 (**D**), FXR (**E**) and SHP (**F**) in the liver of SD rats. The levels of FXR (**G**), FGF-15 (**H**), TGR5 (**I**) and GLP-1 (**J**) in the ileum of SD rats. The levels of PGC1α (**K**) and UCP1 (**L**) in the perirenal adipose tissue for SD rats. Data were denoted as mean ± standard deviation. Significant differences among the four groups were shown with different letters. CYP7A1, cholesterol 7α-hydroxylase; CYP8B1, sterol 12α-hydroxylase; CYP27A1, sterol-27-hydroxylase; CYP7B1, oxysterol 7α-hydroxylase; FXR, farnesoid X receptor; SHP, small heterodimer partner; FGF-15, fibroblast growth factor 15; TGR5, G protein-coupled receptor 5; GLP-1, glucagon-like peptide-1; PGC1α, peroxisome proliferators-activated receptor γ coactivator l-alpha; UCP1, uncoupling protein 1.

**Figure 2 nutrients-15-04929-f002:**
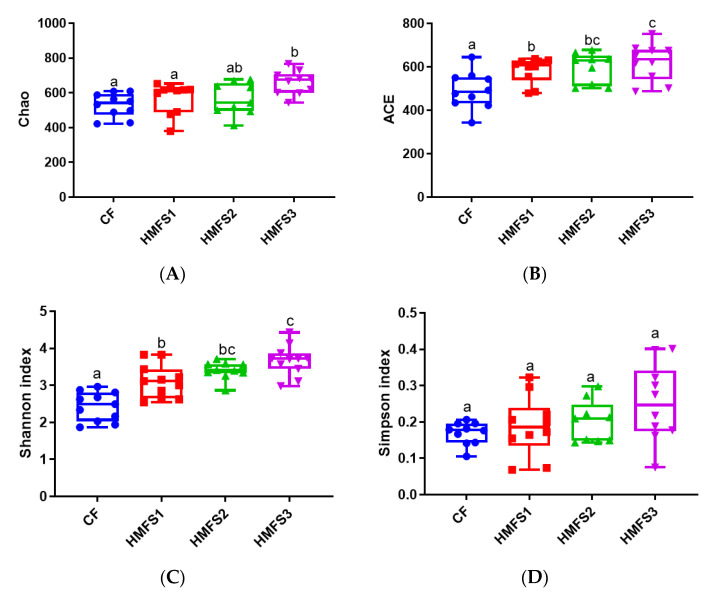
The α-Diversity indices of gut microbiota in SD rats. Chao (**A**), ACE (**B**), Shannon (**C**) and Simpson (**D**) index. Significant differences among the four groups were shown with different letters (*p* < 0.05).

**Figure 3 nutrients-15-04929-f003:**
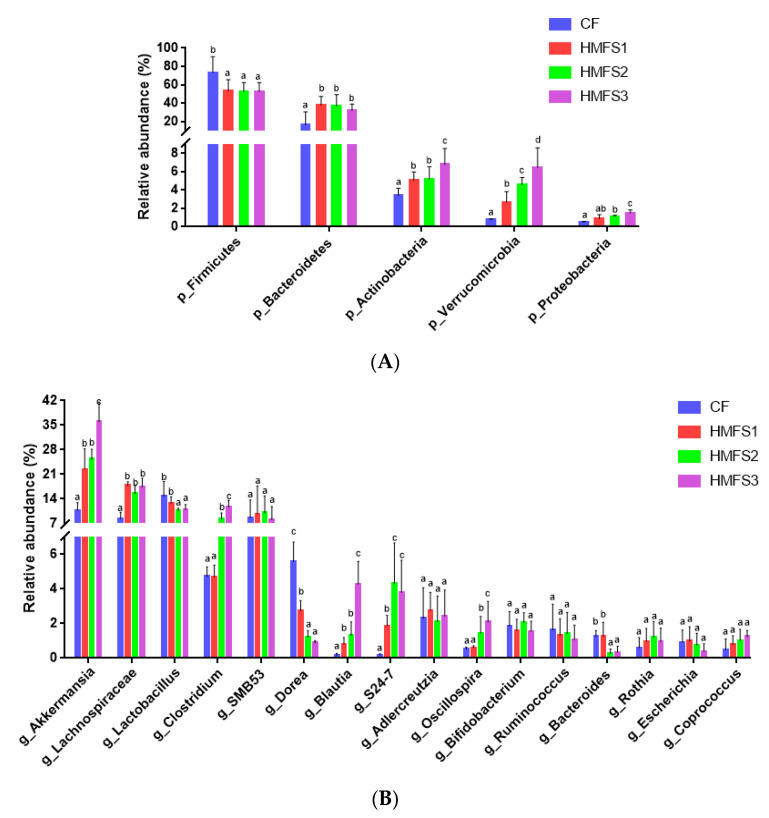
The relative abundance for gut microbiota at the phylum (**A**) and genus (**B**) level in SD rats. The results were denoted as mean ± standard deviation. Significant differences among the four groups were shown with different letters (*p* < 0.05). Phyla and genera with a mean relative abundance <1% and unclassified families are not shown.

**Figure 4 nutrients-15-04929-f004:**
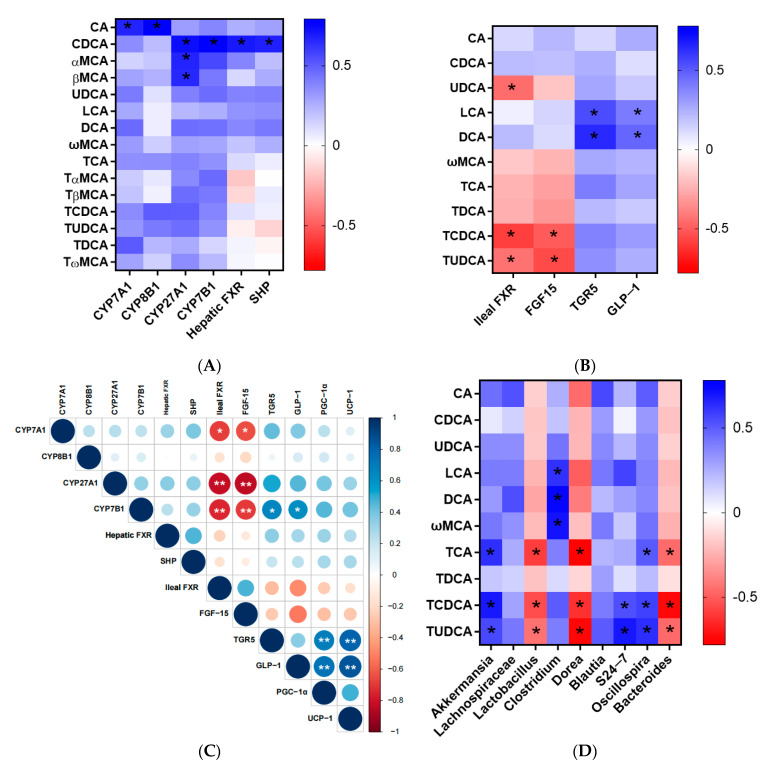
Spearman correlation analysis between hepatic BAs and protein levels (**A**), ileal BAs and protein levels (**B**). Spearman correlation analysis among different protein levels (**C**). The diameter of the spots is proportional to the protein level. Spearman correlation analysis between ileal BAs and relative abundances for gut microbiota (**D**) (* *p* < 0.05, ** *p* < 0.01).

**Figure 5 nutrients-15-04929-f005:**
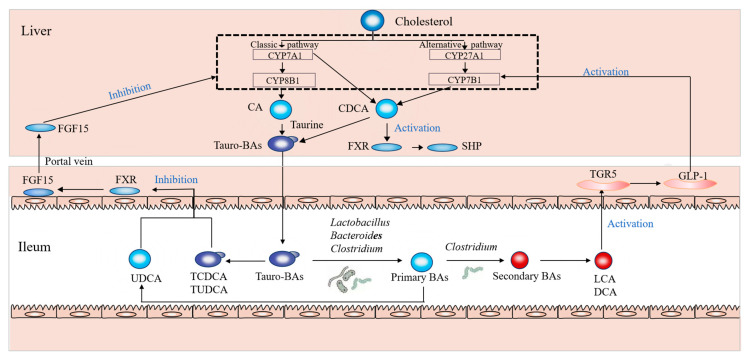
Schematic hypothesis of the interrelation among the main BAs, key proteins involved in BA metabolism and BA-associated gut microbiota.

**Table 1 nutrients-15-04929-t001:** Hepatic BA contents for SD rats (nmol/g).

			CF	HMFS1	HMFS2	HMFS3
Primary BAs	Free BAs	CA	5.65 a ± 0.73	6.81 ^ab^ ± 1.09	7.29 b ± 1.02	7.64 b ± 0.84
		CDCA	2.62 a ± 0.51	5.36 b ± 0.88	7.08 c ± 1.13	10.96 d ± 1.64
		UDCA	5.62 a ± 0.90	5.89 a ± 1.13	6.21 a ± 1.18	6.67 a ± 0.84
		αMCA	1.37 a ± 0.19	3.42 b ± 0.44	3.81 b ± 0.69	6.70 c ± 1.07
		βMCA	2.53 a ± 0.15	6.69 b ± 1.31	7.23 b ± 0.94	10.47 ^c^ ± 1.57
	Tauro-conjugated BAs	TCA	11.65 ^a^ ± 1.51	11.98 ^ab^ ± 1.82	12.29 ^b^ ± 1.29	13.89 ^c^ ± 2.03
		TCDCA	8.54 ^a^ ± 1.11	10.76 ^b^ ± 2.04	11.23 ^c^ ± 0.89	14.35 ^d^ ± 2.44
		TUDCA	8.89 ^a^ ± 1.42	8.96 ^a^ ± 1.85	9.11 ^a^ ± 1.60	9.42 ^a^ ± 2.08
		TαMCA	7.95 a ± 1.03	10.11 b ± 1.52	16.29 c ± 2.77	18.04 d ± 3.60
		TβMCA	4.59 a ± 0.67	7.34 b ± 1.17	11.61 c ± 2.21	16.62 d ± 2.33
Secondary BAs	Free BAs	DCA	0.51 a ± 0.05	0.77 a ± 0.13	1.31 a ± 0.25	1.69 a ± 0.29
		LCA	0.85 a ± 0.17	1.61 a ± 0.27	1.77 a ± 0.35	2.08 a ± 0.29
		ωMCA	1.49 ^a^ ± 0.24	1.62 ^a^ ± 0.21	1.79 ^a^ ± 0.30	1.91 ^a^ ± 0.36
	Tauro-conjugated BAs	TDCA	6.35 ^a^ ± 0.69	6.72 ^a^ ± 0.66	7.11 ^a^ ± 0.72	7.29 ^a^ ± 0.93
		TωMCA	5.61 a ± 0.73	6.18 ab ± 1.17	6.39 b ± 1.06	6.94 b ± 0.83

Data were denoted as mean ± standard deviation. Significant differences among the four groups were shown with different letters. CA, cholic acid; CDCA, chendoxycholic acid; UDCA, ursodeoxycholic acid; αMCA, α muricholic acid; βMCA, β muricholic acid; TCA, tauro cholic acid; TCDCA, tauro chendoxycholic acid; TUDCA, tauro ursodeoxycholic acid; TαMCA, tauro α muricholic acid; TβMCA, tauro β muricholic acid; DCA, deoxycholic acid; LCA, lithocholic acid; ω MCA, ω muricholic acid; TDCA, tauro deoxycholic acid; TωMCA, tauro ω muricholic acid.

**Table 2 nutrients-15-04929-t002:** Ileal BA contents for SD rats (nmol/g).

			CF	HMFS1	HMFS2	HMFS3
Primary BAs	Free BAs	CA	13.13 ^a^ ± 1.46	14.92 ^a^ ± 3.11	16.76 ^a^ ± 2.21	16.92 ^a^ ± 3.29
		CDCA	8.39 ^a^ ± 1.05	8.87 ^a^ ± 1.66	10.73 ^a^ ± 1.69	11.92 ^a^ ± 2.01
		UDCA	3.74 ^a^ ± 0.51	9.20 ^b^ ± 1.76	12.47 ^bc^ ± 2.62	14.82 ^c^ ± 1.98
	Tauro-conjugated BAs	TCA	8.56 ^a^ ± 1.20	16.87 ^b^ ± 1.19	30.96 ^c^ ± 4.49	32.58 ^c^ ± 2.87
		TCDCA	6.65 ^a^ ± 0.99	9.26 ^b^ ± 1.16	11.44 ^b^ ± 2.10	25.70 ^c^ ± 2.49
		TUDCA	3.94 ^a^ ± 0.55	9.15 ^b^ ± 0.62	23.03 ^c^ ± 2.27	34.41 ^d^ ± 5.84
Secondary BAs	Free BAs	DCA	6.65 ^a^ ± 0.99	9.26 ^a^ ± 1.16	16.44 ^b^ ± 3.10	19.72 ^c^ ± 4.49
		LCA	3.94 ^a^ ± 0.55	7.95 ^b^ ± 1.32	11.03 ^c^ ± 1.27	15.41 ^d^ ± 2.84
		ωMCA	11.08 ^a^ ± 1.72	14.78 ^a^ ± 3.17	16.14 ^b^ ± 2.34	19.03 ^b^ ± 3.24
	Tauro-conjugated BAs	TDCA	2.45 ^a^ ± 0.32	8.64 ^ab^ ± 0.39	10.81 ^b^ ± 2.09	11.65 ^b^ ± 2.61

Data were denoted as mean ± standard deviation. Significant differences among the four groups were shown with different letters.

## Data Availability

The data that support the findings of this study are available from the corresponding author upon reasonable request.

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
