# Peer review of "Total Sn-2 Palmitic Triacylglycerols and the Ratio of OPL to OPO in Human Milk Fat Substitute Modulated Bile Acid Metabolism and Intestinal Microbiota Composition in Rats"

_nutrients, 2023, doi:10.3390/nu15234929_

Round 1
Reviewer 1 Report
Comments and Suggestions for Authors
Abstract:
1. The authors tend to use run-on sentences, please cut long sentences into short ones.
2. It is better to add a single sentence at the end that highly summarizes the main finding of this study.
Introduction
1. It is true that no studies like this have been done before, but it does not serve as the rationale for the study. The rationale needs to be elaborated based on the previous findings and the connection between that and the hypothesis.
Methods
1. The authors should either have a list of abbreviations or spell them all out the first time. Otherwise, it reads like a paragraph with code waiting for the readers to decipher.
2. If abbreviations were introduced the first time, there is no need to spell them out again. For example, simply use SD rats instead of Sprague-Dawley (SD) rats in the later paragraphs.
3. Please specify the age of the rats. Newly weaned means 3 weeks?
4. Section 2.5 – how were the ileum and adipose tissue processed prior to measuring using ELISA?
Results and Discussion
1. Please be more specific with the section title, for example, Statistical analyses of XXX
2. I personally prefer a separate Results section and a Discussion section. If the authors think a combined section is better in interpreting the results, it is fine with me. However, the results still need to be reported first and then the discussion.
3. Double-check all figure legends.
Comments on the Quality of English LanguageThe English language is generally good. Minor style and spelling need to be double-checked before publication.
Author Response
1. Reviewer #1:Abstract: The authors tend to use run-on sentences, please cut long sentences into short ones.
Response: “Moreover, increasing sn-2 palmitic TAGs and OPL to OPO ratio in HMFS also altered the microbiota composition both on the phylum and genus level in rats, predominantly microbes associated with bile-salt hydrolase activity, short-chain fatty acid production and reduced obesity risk, suggesting a beneficial effect on host microbial ecosystem.” has been replaced with “Moreover, increasing sn-2 palmitic TAGs and OPL to OPO ratio in HMFS also altered the microbiota composition both on the phylum and genus level in rats, predominantly microbes associated with bile-salt hydrolase activity, short-chain fatty acid (SCFA) production and reduced obesity risk, which suggested a beneficial effect on host microbial ecosystem.” (Line 33-36).
2. Reviewer #1:Abstract: It is better to add a single sentence at the end that highly summarizes the main finding of this study.
Response: Done. “These observations provided important nutritional evidence for the development of new human milk fat substitute products for infants.” has been added at the end of the abstract (Line 37-38).
3. Reviewer #1:Introduction: It is true that no studies like this have been done before, but it does not serve as the rationale for the study. The rationale needs to be elaborated based on the previous findings and the connection between that and the hypothesis.
Response: No changes have been made in the revision. Based on your comments, we have reviewed the Introduction Part carefully. It is found that the connection between previous findings and our hypothesis in this study has already been discussed and elaborated (Line 49-60, 71-73).
4. Reviewer #1:Methods: The authors should either have a list of abbreviations or spell them all out the first time. Otherwise, it reads like a paragraph with code waiting for the readers to decipher.
Response: Done. We have added a list of abbreviations in the Abbreviations Part at the end of the revision (Line 479-523).
5. Reviewer #1:Methods: If abbreviations were introduced the first time, there is no need to spell them out again. For example, simply use SD rats instead of Sprague-Dawley (SD) rats in the later paragraphs.
Response: Done. “Sprague-Dawley” has been replaced with “SD” (Line 76, 168, 186, 202-203, 229, 248, and Line 463-465). Meanwhile, we have deleted the full names of other repeated words, including OPO, OPL, and HMFS in the Introduction Part.
6. Reviewer #1:Methods: Please specify the age of the rats. Newly weaned means 3 weeks?
Response: Done. We have added the age of the rats in the revision (Line 92- 93).
7. Reviewer #1:Methods: Section 2.5 - how were the ileum and adipose tissue processed prior to measuring using ELISA.
Response: Done. “The liver (110 mg), ileum (80 mg) and adipose tissue (50 mg) were rinsed with pre-cooled PBS (pH=7.4) to remove the residual blood, respectively. Meanwhile, 1.1 mL, 0.8 mL and 0.5 mL of the PBS was added to them, respectively. After homogenizing and grounding, the homogenate was centrifuged at 5000 g for 10 min and the supernatant was taken for testing.” has been added at the Section 2.5 to introduce the pre-treatment methods (Line 111-115).
8. Reviewer #1:Results and Discussion: Please be more specific with the section title, for example, Statistical analyses of XXX
Response: Done. “Statistical analyses” has been replaced with “Hepatic and ileal BAs contents” (Line 144).
9. Reviewer #1:Results and Discussion: I personally prefer a separate Results section and a Discussion If the authors think a combined section is better in interpreting the results, it is fine with me. However, the results still need to be reported first and then the discussion.
Response: Done. To better explain the results, we have split the Results and Discussion section into two sections, and added a schematic diagram to interprete the interrelation of all the results obtained in this study (Figure 5, Line 450).
10. Reviewer #1:Results and Discussion: Double-check all figure legends.
Response: Done. We have checked and rewritten all the figure legends.
Reviewer 2 Report
Comments and Suggestions for Authors
It is opinion of the reviewer that this paper before acceptance needs several corrections. My individual comments are listed below.
A new title is requested. Don’t use abbreviations. “Sprague-Dawley” is not important information.
A title with capital letters.
L. 8-11 – The authors’ I e-maail addresses and initials should be completed.
L. 14 – “sn-“ with italic.
L. 15 – Explanation of “OPL” and “OPO” is needed.
L. 53 – “on illustrating”?
L. 58 – It should be “lipids” instead of “fat”.
L. 79/80 – Full name of the bile acids.
L. 113 – The short description of the method should added.
L. 122 – How it was determined.
L. 127 -– It should be “Briefly, the …”.
L. 138 – The model of the used fluorometer.
L. 159 – “in vivo” with italic.
L. 161 – “lipid indicators”?
L. 189 – An unit (nmol/g) remove move from the table to the table title.
L. 191, 244 – Add full name of Bas to the footnote.
L. 199, 201 – The results should be reported with one digital after decimal point.
In tables and on figures the highest value should be marked with “a”, lower with “b”, etc.
L. 339 – It should be “α-Diversity …”.
L. 470-473 – Initials instead of full names.
DOI should be added to references.
L. 514 – It should be “BMC Biol.”
L. 522 – It should be “PLoS ONE”.
L. 595 – It should be “J. Hepatol.”.
Comments on the Quality of English Language
The correction by a Native speaker is not needed.
Author Response
1. Reviewer #2:Title: A new title is requested. Don’ t use abbreviations. “Sprague-Dawley” is not important information.
Response: Done. “Sprague-Dawley” has been deleted from the title. Since OPO and OPL are common triglycerides, it seems not necessary to add their full names in the title. Otherwise, the title could be too long. Therefore, the title has been changed from “Total Sn-2 palmitic triacylglycerols and the ratio of OPL to OPO in human milk fat substitute modulated bile acid metabolism and intestinal microbiota composition in newly-weaned Sprague-Dawley rats” to “Total Sn-2 Palmitic Triacylglycerols and the Ratio of OPL to OPO in Human Milk Fat Substitute Modulated Bile Acid Metabolism and Intestinal Microbiota Composition in Rats” (Line 2-4).
2. Reviewer #2:Title: A title with capital letters.
Response: Done. “Total Sn-2 palmitic triacylglycerols and the ratio of OPL to OPO in human milk fat substitute modulated bile acid metabolism and intestinal microbiota composition in newly-weaned Sprague-Dawley rats” has been replaced with “Total Sn-2 Palmitic Triacylglycerols and the Ratio of OPL to OPO in Human Milk Fat Substitute Modulated Bile Acid Metabolism and Intestinal Microbiota Composition in Rats” (Line 2-4).
3. Reviewer #2:Line 8-11: The authors’ e-mail addresses and initials should be completed.
Response: Done. The e-mail addresses and initials of authors have been added (Line 7-14).
4. Reviewer #2:Line 14: “sn-” with italic.
Response: Done. “sn-” has been replaced with “sn-” in the revision (Line 16).
5. Reviewer #2:Line 15: Explanation of “OPL” and “OPO” is needed.
Response: Done. “OPL to OPO” has been replaced with “1-oleoyl-2-palmitoyl-3-linoleoylglycerol (OPL) to 1,3-dioleoyl-2-palmitoylglycerol (OPO)” (Line 17-18).
6. Reviewer #2:Line 53: “on illustrating” ?
Response: Done. “Therefore, more studies have concentrated on illustrating the effects of dietary fats on host health through interpreting the roles of gut microbiota” has been replaced with “Therefore, more studies have investigated the effects of dietary lipids on host health through interpreting the roles of gut microbiota” (Line 62-63).
7. Reviewer #2:Line 58: It should be “lipids” instead of “fat”.
Response: Done. “fats” has been replaced with “lipids” (Line 44-45, 47, 62 and Line 95).
8. Reviewer #2:Line 79-80: Full name of the bile acids.
Response: Done. To avoid repetition, the abbreviations for BAs have been deleted in the revision. Meanwhile, the first occurrence of cholic acid in the revision has been shown in its full name.
9. Reviewer #2:Line 113: The short description of the method should added.
Response: No changes have been made in the revision. Since this work is a continuation of our previous work, the method was the same with that previously reported (Zhu, L.; Fang, S. Z.; Liu, W. W.; Zhang, H.; Zhang, Y. Q.; Xie, Z. H.; Yang, P. Y.; Wan, J. C.; Gao, B. Y.; Yu, L. L. Triacylglycerol structure and composition of human milk fat substitute affect the absorption of fatty acids and calcium, lipid metabolism and bile acid metabolism in newly-weaned Sprague-Dawley rats. Food Funct. 2023, 14, 7574.). The detailed description of the method is not added in this part to avoid repetition.
10. Reviewer #2:Line 122: How it was determined.
Response: Done. “DNA concentration and purity were obtained by determing the absorption at 260 and 280 nm using the NanoDrop spectrophotometer (Thermo Scientific, Wilmington, USA).” has been added at the Section 2.6 in the revision (Line 122-124).
11. Reviewer #2:Line 127: It should be “Briefly, the ...”.
Response: Done. “Briefly, The” has been replaced with “Briefly, the” (Line 126).
12. Reviewer #2:Line 138: The model of the used fluorometer.
Response: Done. The model of the used fluorometer is E6150. Meanwhile, “Quantus Fluorometer E6150 (Promega, USA) was used to quantify the product. ” has been rewritten at the Section 2.7 in the revision (Line 128-129).
13. Reviewer #2:Line 159: “in vivo” with italic.
Response: Done. “in vivo” has been replaced with “in vivo” (Line 273).
14. Reviewer #2:Line 161: “lipid indicators”?
Response: No changes have been made in the revision. Serum lipid indicators refer to four serum parameters, that is total triacylglycerol, total cholesterol, low-density lipoprotein-cholesterol and high-density lipoprotein-cholesterol (Line 275-277).
15. Reviewer #2:Line 189: An unit (nmol/g) remove move from the table to the table title.
Response: Done. The unit (nmol/g) has been moved to the table title (Table 1 and 2, Line 168 and 186).
16. Reviewer #2:Line 191, 244: Add full name of BAs to the footnote.
Response: Done. We have added a list of abbreviations in the Abbreviations Part at the end of the revision (Line 479-523). Meanwhile, “The full names of BAs shown in the table are listed in the Abbreviations Part.” has been added to the footnote of Table 1 and 2 (Line 171-172, 189-190).“The full names of proteins shown in the figures are listed in the Abbreviations Part.” has been added to the footnote of Figure 1 (Line 205-206). “The full names of BAs and proteins are listed in the Abbreviations Part.” has been added to the footnote of Figure 4 (Line 271).
17. Reviewer #2:Line 199, 201: The results should be reported with one digital after decimal In tables and on figures the highest value should be marked with “a”, lower with “b”, etc.
Response: Done. The data in Table S1 has all been shown with one digital after decimal point.
Besides, based on several previously literatures published in Nutrients, the lowest values could be marked with “a” and the higher values with “b”, etc. (Sánchez-Moreno, C. Bile Acids and Short-Chain Fatty Acids Are Modulated after Onion and Apple Consumption in Obese Zucker Rats. Nutrients, 2023, 15, 3035-3055; Wang, C.; Li, S.; Sun, E.; Xiao, R.; Wang, R.; Ren, Y.; Zhan, J. Effects of fermented milk containing Bifidobacterium animalis Subsp. lactis MN-Gup (MN-Gup) and MN-Gup-based synbiotics on obesity induced by high fat diet in rats. Nutrients, 2022, 14, 2631-2646.). Therefore, no change has been made in the revision.
18. Reviewer #2:Line 339: It should be “α-Diversity ...”.
Response: Done. “α-diversity” has been replaced with “α-Diversity” (Line 218).
19. Reviewer #2:Line 470-473: Initials instead of full names.
Response: Done. The full names have been replaced with initials (Line 467-469).
20. Reviewer #2:References: DOI should be added to references.
Response: Done. DOI has been added to all the references.
21.Reviewer #2:Line 514: It should be “BMC Biol.”.
Response: Done. “BMC biol.” has been replaced with “BMC Biol.” (Line 555).
22. Reviewer #2:Line 522: It should be “PLoS ONE”.
Response: Done. “PLoS One” has been replaced with “PLoS ONE” (Line 563).
23. Reviewer #2:Line 595: It should be “ Hepatol.”.
Response: Done. “J. hepatol.” has been replaced with “J. Hepatol.” (Line 639).
Reviewer 3 Report
Comments and Suggestions for Authors
The manuscript untitled “Total Sn-2 palmitic triacylglycerols and the ratio of OPL to OPO in human milk fat substitute modulated bile acid metabolism and intestinal microbiota composition in newly-weaned Sprague-Dawley rats” describes part of the results of a larger work on the effect of structured triglycerides on lipid metabolism.
The results described are interesting. However, the biggest problem I find in the manuscript is a lack of cohesion between the results and their discussion and interpretation.
The Increase of sn-2 palmitic acid level and OPL to OPO ratio caused changes in bile acids synthesis, related to the level of enzymes involved in their synthesis, both in liver and in ileum. Also they described differences in the expression of some factors that participate in the regulation.
And finally, they described changes found in gut microbiota.
However, after reading the entire manuscript the feeling is that I don't quite see the interrelation between all these aspects. And above all, I have not seen in the discussion any hypothesis that explains why these structured triglycerides can produce these effects.
In my opinion, the authors devote too much space to the detailed description of the results and little to their discussion. Furthermore, at some points the description is too detailed and difficult to follow.
I would suggest separating the results from the discussion. Since the results are seen very clearly in the tables and figures provided, I would suggest greatly summarize their description and include a discussion section apart.
The discussion should be focused on explaining the interrelationship between all the results obtained and on trying to explain the role that structured triglycerides play in them. Especially considering that this is the objective of the study and, in my opinion, is has not been achieved.
I also think that the authors should explain why they use a high-fat diet, especially considering that the ultimate application could be infant feeding. In my opinion, they should have included at least a normal fat control.
More especific comments
The meaning of the abbreviations of all names of bile acids and enzymes and proteins should be included the first time they are mentioned in the text, both in the abstract and in the main text. Please review this aspect throughout the manuscript.
The expression "protein expression" in my opinion is not correct. Genes are expressed to produce proteins. Therefore, the level (or concentration) of enzymes and protein in the different samples is measured, but not their expression. Please review also this aspect throughout the manuscript.
Abstract
Please, define all abbreviations.
Line 23. As far as I know there is no enzyme called BA synthetase.
Introduction
Line 57. The intestinal microbiota does not produce bile acids, but rather modifies them.
M & M
Section 2.5
As I said previuosly, "gene expression" has protein as product and with ELISA kits authors are measuring protein concentration (not protein expression).
All this paragraph should be rewritten taking this into consideration.
In addition, the meaning of abbreviations must be included
Results and Discussion
The manuscript would be greatly improved by separating the results and discussion sections. Additionally, the description of the results should be summarized, since some paragraphs are difficult to follow.
Conclusions
I do not agree with the statement “this study showed that sn-2 palmitic TAGs and the OPL to OPO ratio human milk fat substitute significantly influenced bile acid metabolism through FXR and TGR5 mediated signaling pathways in newly-weaned Sprague-Dawley (SD) rats. …..
At most they could say that there seems to be a significant relationship between the factors studied. But, as I have mentioned, they do not demonstrate that structured triglycerides have a direct effect (that they are the cause) of the results obtained. Nor do they propose a hypothesis that could explain this relationship.
Other minor comments are included in the revised manuscript

Comments on the Quality of English LanguageIn general, the English is correct although it has room for improvement
Author Response
1.Reviewer #3:However, the biggest problem I find in the manuscript is a lack of cohesion between the results and their discussion and interpretation. However, after reading the entire manuscript the feeling is that I don't quite see the interrelation between all these aspects. And above all, I have not seen in the discussion any hypothesis that explains why these structured triglycerides can produce these effects.
Response: Done. To clarify the interrelation among all the obtained results, results and discussion has been separated into two parts in the revision. Meanwhile, a schematic diagram for elaborating the interrelation of the main results in this study is also added (Figure 5) (Line 450).
In this study, the influence of sn-2 palmitic TAGs and OPL to OPO ratio in HMFS on the BA profiles, BA metabolsim and gut microbiota has been observed in rats. As for the reason behind these observations, we are still working on it. It might be due to the different absorption efficiency of sn-2 palmitic TAGs. In a recently published cellular study, it has been found that OPL has a higher absorption efficiency compared with that of OPO (Zhang, N.; Zeng, J. P.; Wu, Y. P.; Wei, M.; Zhang, H.; Zheng, L.; Li, J. Human milk sn-2 palmitate triglyceride rich in linoleic acid had lower digestibility but higher absorptivity compared with the sn-2 palmitate triglyceride rich in oleic acid in vitro. Journal of Agricultural and Food Chemistry, 2020, 69, 9137-9146).
2. Reviewer #3:In my opinion, the authors devote too much space to the detailed description of the results and little to their discussion. Furthermore, at some points the description is too detailed and difficult to follow. I would suggest separating the results from the discussion. Since the results are seen very clearly in the tables and figures provided, I would suggest greatly summarize their description and include a discussion section apart. The discussion should be focused on explaining the interrelationship between all the results obtained and on trying to explain the role that structured triglycerides play in them. Especially considering that this is the objective of the study and, in my opinion, is has not been achieved.
Response: Done. According to the reviewer’s comments, we have separated the results from the discussion in the revision. Meanwhile, a schematic diagram for elaborating the interrelationship of the main results in this study is also added (Figure 5) (Line 450).
3.Reviewer #3:I also think that the authors should explain why they use a high-fat diet, especially considering that the ultimate application could be infant feeding. In my opinion, they should have included at least a normal fat control.
Response: The fat content in diets used in this study is 15 wt% (Table S2), which is similar to that in infant formula containing fat content of 22% (Happe, R. P.; Gambelli, L. Infant formula. In Specialty oils and fats in food and nutrition. Woodhead Publishing, 2015, 285-315.). Compared to that of adults, infants have higher energy needs for growth and development.
4. Reviewer #3:The meaning of the abbreviations of all names of bile acids and enzymes and proteins should be included the first time they are mentioned in the text, both in the abstract and in the main text. Please review this aspect throughout the manuscript.
Response: Done. We have reviewed the manuscript carefully. The full names of BAs and proteins have been added for the first time mentioned in the revision.
5.Reviewer #3:The expression “protein expression” in my opinion is not correct. Genes are expressed to produce proteins. Therefore, the level (or concentration) of enzymes and protein in the different samples is measured, but not their expression. Please review also this aspect throughout the manuscript.
Response: Done. “Protein expression” has been replaced with “protein level” throughout the manuscript (Line 110, 174, 194, 213, 268-270 and Line 297).
6.Reviewer #3:Abstract: Please, define all abbreviations.
Response: Done. The full names of all abbreviations have been added in the abstract (Line 17-18, 21-28, 30-32 and Line 35).
7.Reviewer #3:Line 23: As far as I know there is no enzyme called BA synthetase.
Response: Done. “BA synthetase” has been replaced with “enzymes involved in BA synthesis” (Line 29-30).
8.Reviewer #3:Line 57: The intestinal microbiota does not produce bile acids, but rather modifies them.
Response: Done. To avoid repetition, the “One important class of microbially produced metabolites is bile acids (BAs), which are essential for the digestion and absorption of dietary fats in the host.” has been deleted.
9. Reviewer #3:Section 2.5: As I said previuosly, “gene expression” has protein as product and with ELISA kits authors are measuring protein concentration (not protein expression). All this paragraph should be rewritten taking this into consideration.
Response: Done. “Protein expression” has been replaced with “protein level” throughout the manuscript (Line 110).
10. Reviewer #3:Section 2.5: In addition, the meaning of abbreviations must be included.
Response: Done. The full names of proteins have been added in Section 2.5 (Line 115-116).
11. Reviewer #3:Results and Discussion: The manuscript would be greatly improved by separating the results and discussion sections. Additionally, the description of the results should be summarized, since some paragraphs are difficult to follow.
Response: Done. We have separated the results from the discussion in the revision. Meanwhile, a schematic diagram for elaborating the interrelationship of the main results in this study is also added (Figure 5) (Line 450).
12. Reviewer #3:Conclusions: I do not agree with the statement “this study showed that sn-2 palmitic TAGs and the OPL to OPO ratio human milk fat substitute significantly influenced bile acid metabolism through FXR and TGR5 mediated signaling pathways in newly-weaned Sprague-Dawley (SD) rats. …..At most they could say that there seems to be a significant relationship between the factors studied. But, as I have mentioned, they do not demonstrate that structured triglycerides have a direct effect (that they are the cause) of the results obtained. Nor do they propose a hypothesis that could explain this relationship.
Response: Done. “In conclusion, this study showed that sn-2 palmitic TAGs and the OPL to OPO ratio in human milk fat substitute significantly influenced bile acid metabolism through FXR and TGR5 mediated signaling pathways in newly-weaned SD rats.” has been replaced with “In conclusion, this study showed that sn-2 palmitic TAGs and the OPL to OPO ratio in human milk fat substitute had some influence on the BA metabolism through FXR and TGR5 mediated signaling pathways in newly-weaned SD rats.” (Line 455).
13.Reviewer #3:Line 2: “Sn” ?
Response: No changes have been made in the revision. Since the title needs to be capitalized according to Nutrients.
14.Reviewer #3:Line 18-20: Abbreviations sholud be defined.
Response: Done. The full names of all abbreviations have been added in the abstract (Line 17-18, 21-28, 30-32 and Line 35).
15.Reviewer #3:Line 23:“BA synthetase”?
Response: Done. “Increased expressions of bile acid synthetase” has been replaced with “increased levels of enzymes involved in BA synthesis” (Line 29-30 ).
16.Reviewer #3:Line 24: It should be “genes” instead of “proteins”.
Response: No changes have been made in the revision. It’s true that the levels of two key thermogenic proteins (PGC1α and UCP1) has been measured in this manuscript.
17.Reviewer #3:Line 29: “Sn” lower case.
Response: Done. “Sn-2 palmitic triacylglycerols” has been replaced with “sn-2 palmitic triacylglycerols” (Line 39).
18.Reviewer #3:Line 33: It should be “the main components”
Response: Done. “Fatty acids are important components of triacylglycerols (TAGs), which are the major components of dietary fats.” has been replaced with “Fatty acids are the main components of TAGs, which are the major components of dietary lipids.” (Line 43-44).
19. Reviewer #3:Line 56-58: “One important class of microbially produced metabolites is bile acids (BAs), which are essential for the digestion and absorption of dietary fats in the host”? The intestinal microbiota does not produce bile acids, but rather modifies them.
Response: Done. To avoid repetition, the “One important class of microbially produced metabolites is bile acids (BAs), which are essential for the digestion and absorption of dietary fats in the host.” has been deleted.
20.Reviewer #3:Line 65-67: “To the best of our knowledge, no relevant in vivo studies have been reported on whether and how sn-2 palmitic TAGs and the ratio of OPL to OPO may alter on the BA metabolism and gut microbiota composition”. Why should them have any influence?
Response: No changes have been made in the revision. Our previous studies have shown that both sn-2 palmitic TAGs and OPL to OPO ratio in HMFS could affect the lipid metabolism in rats. Meanwhile,it is known that the lipid metabolism, BA metabolism and gut microbiota are interrelated. Therefore, the influence of sn-2 palmitic TAGs and OPL to OPO ratio on the BA metabolism and gut microbiota is investigated in this study. The purpose for this study has already been discussed and elaborated in the Introduction Part (Line 54-60, 61-73).
21.Reviewer #3:Line 73: “key protein expressions”?
Response: Done. “Key protein expressions of” has been replaced with “key proteins involved in” (Line 79).
22. Reviewer #3:Line 79-80: Definition of abbrevitions.
Response: Done. The full names of abbreviations the first time mentioned have been added in the Section 2.5. (Line 115-116).
23. Reviewer #3:Line 114, 217, 224, 225, 231, 255, 309, 310: The “expression” should be deleted.
Response: Done. The “expression” has been deleted.
24.Reviewer #3:Line 158: “Statistical analyses”?
Response: Done. “Statistical analyses” has been replaced with “Hepatic and ileal BAs contents” (Line 144).
25.Reviewer #3:Results and Discussion: Double-check all figure legends.
Response: Done. We have checked and rewritten all the figure legends.
26.Reviewer #3:Line 206-208: Data are in the table, is not necsesary to repeat.
Response: Done. “(19.72 nmol/g and 19.03 nmol/g), compared to that of HMFS1-fed (9.26 nmol/g and 14.78 nmol/g) and HMFS2-fed rats (16.44 nmol/g and 16.14 nmol/g)” has been deleted.
27. Reviewer #3:Line 236: This enzyme- BA synthase does not exist.
Response: Done. “BA synthetase” has been replaced with “enzymes involved in BA synthesis” (Line 320).
28.Reviewer #3:Line 361: “decreased”?
Response: Done. “Decreased” has been deleted.
Round 2
Reviewer 3 Report
Comments and Suggestions for Authors
The authors have satisfactorily answered most of the questions raised. Thus, comparing with the previuos version, the manuscript has improved considerably, especially, refering to results description.
However, in the discussion authors make the same mistake that they did in the previous version with the results. They include a too detailled description of all the positive relationships between parameters they have found, making the discussion difficult to follow. First of all, they are result clearly shown in figure 4. So, in any case, they must be described in the results section. However, as I said, they can be clearly seen in Figure 4.
Furthermore, some correlations are so obvious (e.g., the level of BA and the enzymes that synthesize them) that I do not see the need to comment on them.
On the other hand, I find the scheme they have included, explaining the hypothesis on the effect of structured triglycerides, very interesting and clarifying. Therefore, I would advise the authors to reduce the discussion and focus it on discussing the results that support this hypothesis.
In the discussion they, quite often, use the term “synergistically reduce” or “synergistically increase”. I don't quite understand what they mean by this. In any case, they do not demonstrate that it is the structured triglyceride that produces the increase or decrease in the parameters studied. The only thing they demonstrate is that there is a significant relationship between them.
If I understand correctly, the scheme in figure 5 is an original proposal, from the authors, to explain the results they have obtained. If so, they should make it clear both in the figure caption and in the text. It is an original proposal, from the authors, to explain the results they have obtained. If not, they must cite the bibliography on which they have based it.
They included the meaning of most the abbreviations only in the abstract. But, as it is stated in “instructions for authors “Abbreviations/Initialisms should be defined the first time they appear in each of three sections: the abstract; the main text; the first figure or table.
I actually don't see the need to include most abbreviations in the abstract if the full term is mentioned only once in it.
Conclusions: Taking into account the hypothesisi in figure 5, in my opinion, it would be more correct to say that that sn-2 palmitic TAGs and the OPL to OPO ratio 454 in human milk fat substitute had influence in the gut microbiota that, in turn, can affect BA metabolism through ......

Comments on the Quality of English LanguageModerate English Editing is Needed.
Author Response
Itemized List of the Responses to Reviewers’ Comments:
Manuscript ID: nutrients-2689762
Title: Total Sn-2 Palmitic Triacylglycerols and the Ratio of OPL to OPO in Human Milk Fat Substitute Modulated Bile Acid Metabolism and Intestinal Microbiota Composition in Rats
Authors: Lin Zhu, Shuaizhen Fang, Hong Zhang, Xiangjun Sun, Puyu Yang, Jianchun Wan, Yaqiong Zhang*, Weiying Lu, Liangli (Lucy) Yu
1. Reviewer #3:Line 323-325, 332-333, 337-338, 354-356, 362-364, 413-416, and Line 437-439: They include a too detailed description of all the positive relationships between parameters they have found, making the discussion difficult to follow. So, in any case, they must be described in the results section.
Response: Done. The description of all the correlations between parameters has been removed from the Discussion section to the Results section. (Section 3.4, Line 279-318).
2. Reviewer #3:Some correlations are so obvious (e.g., the level of BA and the enzymes that synthesize them) that I do not see the need to comment on them.
Response: No changes have been made in the revision. It is found that not all the hepatic BAs are significantly correlated with the BA synthesis enzymes, except for CA and CDCA. The correlations between CA or CDCA and the specific BA synthesis enzymes could help us understand the regulatory effect of sn-2 palmitic TAGs and OPL to OPO ratio in HMFS on two BA synthesis pathways in rats (the classical and alternative pathways) (Line 336-350).
3. Reviewer #3:I find the scheme they have included, explaining the hypothesis on the effect of structured triglycerides, very interesting and clarifying. I would advise the authors to reduce the discussion and focus it on discussing the results that support this hypothesis.
Response: Done. The discussion part has been re-written (Line 320-429), and the schematic diagram has also been re-organized to be more clarifying (Figure 5).
4. Reviewer #3:Line 285-287 and Line 294-296: “The results showed that increasing the sn-2 palmitic TAGs and OPL to OPO ratio could synergistically increase the contents of 6 primary BAs in the liver.” and“It was also shown that increasing the sn-2 palmitic TAGs and OPL to OPO ratio could synergistically increase the levels of ileal BAs, especially for the tauro-conjugated and secondary BAs.” In the discussion they, quite often, use the term “synergistically reduce” or “synergistically increase”. I don't quite understand what they mean by this. In any case, they do not demonstrate that it is the structured triglyceride that produces the increase or decrease in the parameters studied. The only thing they demonstrate is that there is a significant relationship between them.
Response: Done. “Synergistically” has been replaced with “significantly” throughout the manuscript (Line 152-154 and Line 158-163).
5.Reviewer#3:They should make it clear both in the figure caption and in the text. It is an original proposal, from the authors, to explain the results they have obtained.
Response: Done. “To clarify the influence of sn-2 palmitic TAGs and OPL to OPO ratio in HMFS on the BA metabolsim and gut microbiota more clearly, a schematic diagram outlining the interrelation among the main BAs, key proteins involved in BA metabolism and BA-associated gut microbiota in rats was shown in Figure 5, which is an original hypothesis based on all the results in this study.” has been added in the discussion part (Line 321-324).
6. Reviewer #3:Line 47-49 and Line 54: As it is stated in “instructions for authors”, “Abbreviations/Initialism should be defined the first time they appear in each of three sections: the abstract; the main text; the first figure or table. I actually don't see the need to include most abbreviations in the abstract if the full term is mentioned only once in it.
Response: Done. The full names of the abbreviations were added to the abstract, the main text, and the footnote of the first table or figure sections (Line 37, 43-44, 49, 56, 61-62, 71, 75-76, 81-86, 116-120, 173-177, and Line 208-212). To meet the word count requirement for the abstract in Nutrients, the full names of BA and proteins involved in BA metabolism are not included in the abstract, but they are shown in the Materials and Methods Part.
7. Reviewer #3:Line 455-456: “In conclusion, this study showed that sn-2 palmitic TAGs and the OPL to OPO ratio in human milk fat substitute had some influence on the BA metabolism through FXR and TGR5 mediated signaling pathways in newly-weaned SD rats.” In my opinion, it would be more correct to say that sn-2 palmitic TAGs and the ratio of OPL to OPO in human milk fat substitute had influence in the gut microbiota that, in turn, can affect BA metabolism...
Response: No changes have been made in the revision. Actually, the hepatic and ileal BA profiles, the enzymes involved in the BA metabolism and gut microbiota are interrelated, which has been stated in the Introduction Part (Line 56-64). It is hard to say which one changes first, and then affects another.
8. Reviewer #3:Line 112-113: “The liver (110 mg), ileum (80 mg) and adipose tissue (50 mg) were rinsed with pre-cooled PBS (pH=7.4) to remove the residual blood, respectively. Meanwhile, 1.1 mL, 0.8 mL and 0.5 mL of the PBS was added to them, respectively. After homogenizing and grounding, the homogenate was centrifuged at 5000 g for 10 min and the supernatant was taken for testing.” are bad constructed sentences. Please, correct them.
Response: Done. “The liver (110 mg), ileum (80 mg) and adipose tissue (50 mg) were rinsed with pre-cooled PBS (pH=7.4) to remove the residual blood, respectively. Meanwhile, 1.1 mL, 0.8 mL and 0.5 mL of the PBS was added to them, respectively. After homogenizing and grounding, the homogenate was centrifuged at 5000 g for 10 min and the supernatant was taken for testing.” has been replaced with “The liver (110 mg), ileum (80 mg) and adipose tissue (50 mg) were rinsed with pre-cooled PBS (pH=7.4) to remove the residual blood, respectively. Then, the tissue was homogenized and centrifuged at 5000 g for 10 min in pre-cooled PBS, and the supernatant was taken for testing.” (Line 113-116).
9. Reviewer #3:Line 119: “Respectively” should be deleted.
Response: Done. “Respectively” has been deleted (Line 121).
10. Reviewer #3:Line 174: “2. Protein levels related to BA metabolism” should be italic.
Response: Done. “3.2. Protein levels related to BA metabolism” has been replaced with “3.2. Protein levels related to BA metabolism” (Line 178).
11. Reviewer #3:Line 193: “Expressions” should be deleted.
Response: Done. “Expressions” has been deleted (Line 195-196).
12. Reviewer #3:Line 276: “Triacylglycerol” sholud be abbreviated.
Response: Done. “Triacylglycerol” has been replaced with “TAG” (Line 320).
13. Reviewer #3:Line 357: “αMCA and βMCA are two primary BAs synthesized from CDCA via hydroxylated isomerases in the liver.” hydroxylated isomerases?
Response: Done. Hydroxylated isomerases have been deleted in the revision.
14. Reviewer #3:Line 358-359: “Since the CDCA content was positively correlated with CYP27A1, it was also reasonable that αMCA and βMCA have a similar correlation. ” ?
Response: Done. This sentence has been deleted.
15. Reviewer #3:Line 451: “Figure 5. Schematic diagram of the interrelation among the main BAs, key proteins involved in BA metabolism and BA-associated gut microbiota.” should be “Figure 5. Schematic diagram hypothesis of the interrelation among the main BAs, key proteins involved in BA metabolism and BA-associated gut microbiota.”
Response: Done. “Figure 5. Schematic diagram of the interrelation among the main BAs, key proteins involved in BA metabolism and BA-associated gut microbiota.” has been changed to “Figure 5. Schematic hypothesis of the interrelation among the main BAs, key proteins involved in BA metabolism and BA-associated gut microbiota.” (Line 432-433).
